# LFIR-YOLO: Lightweight Model for Infrared Vehicle and Pedestrian Detection

**DOI:** 10.3390/s24206609

**Published:** 2024-10-14

**Authors:** Quan Wang, Fengyuan Liu, Yi Cao, Farhan Ullah, Muxiong Zhou

**Affiliations:** 1School of Internet of Things Engineering, Wuxi University, Wuxi 214105, China; wangquan@cwxu.edu.cn (Q.W.); farhan.marwat@gmail.com (F.U.); 2School of Computer Science, Nanjing University of Information Science & Technology, Nanjing 210044, China; brise6087@163.com; 3Key Laboratory of Ministry of Public Security for Road Traffic Safety, Traffic Management Research Institute of the Ministry of Public Security, Wuxi 214151, China; mxzhou2007@126.com

**Keywords:** infrared imaging detection, traffic scenes, C2f DWR, CGA Fusion, lightweight head, loss optimization

## Abstract

The complexity of urban road scenes at night and the inadequacy of visible light imaging in such conditions pose significant challenges. To address the issues of insufficient color information, texture detail, and low spatial resolution in infrared imagery, we propose an enhanced infrared detection model called LFIR-YOLO, which is built upon the YOLOv8 architecture. The primary goal is to improve the accuracy of infrared target detection in nighttime traffic scenarios while meeting practical deployment requirements. First, to address challenges such as limited contrast and occlusion noise in infrared images, the C2f module in the high-level backbone network is augmented with a **Dilation-wise Residual (DWR)** module, incorporating multi-scale infrared contextual information to enhance feature extraction capabilities. Secondly, at the neck of the network, a **Content-guided Attention (CGA)** mechanism is applied to fuse features and re-modulate both initial and advanced features, catering to the low signal-to-noise ratio and sparse detail features characteristic of infrared images. Third, a shared convolution strategy is employed in the detection head, replacing the decoupled head strategy and utilizing shared **Detail Enhancement Convolution (DEConv)** and **Group Norm (GN)** operations to achieve lightweight yet precise improvements. Finally, loss functions, **PIoU v2** and **Adaptive Threshold Focal Loss (ATFL),** are integrated into the model to better decouple infrared targets from the background and to enhance convergence speed. The experimental results on the FLIR and multispectral datasets show that the proposed LFIR-YOLO model achieves an improvement in detection accuracy of 4.3% and 2.6%, respectively, compared to the YOLOv8 model. Furthermore, the model demonstrates a reduction in parameters and computational complexity by 15.5% and 34%, respectively, enhancing its suitability for real-time deployment on resource-constrained edge devices.

## 1. Introduction

Image-based target detection technology is becoming increasingly important in various application domains, such as autonomous driving, healthcare, and industrial automation. However, current research on autonomous driving detection predominantly focuses on the visible spectrum (RGB) domain [1], where images are susceptible to environmental and lighting conditions. Detection performance can significantly deteriorate, particularly in low-light and sun-glare environments [2]. Although infrared images, utilizing thermal radiation technology, offer significant advantages in such conditions and effectively compensate for the limitations of visible spectrum images, their practical application still faces numerous challenges. These challenges primarily stem from the characteristics of infrared sensors and the variability of environmental and target conditions. More specifically, infrared images—especially thermal images—are often affected by atmospheric scattering and clutter interference, typically lack texture information, have low resolution, and exhibit high noise levels [3]. These issues substantially impact the performance of infrared sensing systems in target detection tasks and represent technical difficulties that researchers must thoroughly address and overcome [4].

With the development of autonomous driving technology, the demand for Advanced Driver Assistance Systems (ADASs) is growing rapidly, and infrared cameras’ application prospects in the automotive industry are also expanding [5]. To accelerate the adoption of ADAS technology, governments around the world are actively promoting the implementation of relevant policies. For example, the US Department of Transportation’s National Highway Traffic Safety Administration (NHTSA) has issued federal policies related to Highly Automated Vehicles (HAVs), encouraging the broader implementation of infrared cameras in autonomous driving systems [6]. In 2021, FLIR Corporation incorporated infrared cameras into Level 4 autonomous vehicles, while its partner, Veoneer, utilized the same technology to detect pedestrians within a range of 200 m and generate 360-degree panoramic images [7]. In 2019, a total of 4609 passenger vehicles equipped with night vision systems were sold in China, including models such as the Cadillac XT5, XT6, and Hongqi H7, representing an annual growth rate of 65.6%. In the industrial sector, infrared thermal imaging is widely used for predictive maintenance. According to the Federal Energy Management Program (FEMP) [8], using infrared cameras can save 30–40% in maintenance costs, reduce downtime by 35–45%, and eliminate 70–75% of failures. This makes infrared cameras a critical tool for identifying equipment issues, helping to prevent equipment downtime and damage [9].

Deep learning has emerged as a transformative approach in the field of artificial intelligence, characterized by its ability to automatically learn representations from vast amounts of data [10,11]. By utilizing neural networks with multiple layers, deep learning models can capture intricate patterns and complex relationships within the data, enabling advancements in various domains such as image recognition [12], natural language processing [13,14], and speech analysis [15]. In the context of infrared detection for traffic scenarios, deep learning-based detection algorithms can be broadly categorized into two primary types based on their detection stages—two-stage and one-stage methods. Two-stage algorithms, such as R-CNN [16] and Faster R-CNN [17], first employ a Region Proposal Network (RPN) to generate candidate bounding boxes, followed by the classification and position refinement of these proposals. While this approach is known for its high accuracy, it often comes at the cost of slower processing speeds. Herrmann et al. [18] introduced a CNN classifier specifically designed for the fast classification of hot spots in low-resolution Long-Wave Infrared (LWIR) imagery, distinguishing between pedestrians and other thermal sources. Ullah et al. [19] proposed enhancements to Fast R-CNN for infrared pedestrian detection by introducing two variations—Fast R-CNN type 2, which includes an additional convolutional layer to improve detection accuracy, and Fast R-CNN type 3, which reduces input channels to increase processing speed. Galarza-Bravo et al. [20] proposed a novel Faster R-CNN architecture optimized for pedestrian detection in nighttime conditions, leveraging two region proposal networks to facilitate multi-scale detection.

Conversely, one-stage algorithms, such as SSD [21] and the YOLO series, directly predict class probabilities and bounding box coordinates without the need for a proposal generation step, thereby significantly enhancing detection speed, albeit with a marginal trade-off in accuracy. Among these, the YOLO series has gained prominence for its exceptional efficiency and accuracy, achieving widespread success in various detection tasks. YOLOv3 [22] introduced notable improvements by integrating a Feature Pyramid Network (FPN) and incorporating additional anchor boxes, significantly enhancing detection accuracy. YOLOv4 [23] further optimized the network architecture and training strategies, leading to better performance.

A number of YOLO variants have been specifically developed to address the unique challenges of infrared image detection. Ghenescu et al. [24] proposed a YOLO-DarkNet-based approach utilizing LWIR images for the classification of vehicles, people, boats, and animals. Li et al. [25] introduced an improved infrared object detection network, YOLO-FIRI, which enhances accuracy by expanding the shallow network, integrating attention mechanisms, and employing multi-scale detection strategies. Similarly, Luo et al. [26] presented the YOLO-IR model, which improves detection accuracy while reducing computational complexity by incorporating small object detection, a novel focal loss function, and optimized feature extraction operations, making it suitable for deployment in resource-constrained environments. Further advancements have been made by Lu et al. [27], who proposed YOLO-IR-Free, an anchor-free variation of YOLOv7 designed for real-time infrared vehicle detection. This model improves both accuracy and speed while addressing the challenges posed by low contrast and small object detection. Additionally, Cheng et al. [28] enhanced YOLOv5s for nighttime infrared waterborne detection, refining both accuracy and speed through the use of specialized data and optimized detection techniques.

While the aforementioned YOLO variants exhibit superior real-time performance and lightweight characteristics compared to two-stage models, they still encounter notable challenges in resource-constrained environments. These include a high computational burden, reduced accuracy, elevated false alarm rates, and difficulty extracting features from complex backgrounds [29]. To mitigate these limitations, we introduced several enhancements to the YOLOv8 architecture [30]. Specifically, we optimized the backbone and neck components of the network to improve feature extraction and fusion capabilities, incorporated a lightweight detection head to reduce computational complexity, and replaced the loss function to boost detection performance and efficiency. The key contributions of the proposed model are as follows:The fixed nature and lack of flexibility of the standard convolutional kernels in the CSP Bottleneck with two convolutions (C2f) backbone networks limit the receptive field, making it challenging to capture global contextual information. This limitation can result in inaccurate differentiation between targets and backgrounds in low-contrast and low-resolution infrared images, leading to false and missed detections. We introduced the DWR module to address this issue; it leverages dilated convolutions and an efficient residual structure design. This module expands the receptive field while maintaining resolution, enabling efficient and flexible multi-scale feature extraction. By integrating the bottleneck part of C2f with the DWR module, we created a new C2f_DWR module.Low-level features play a crucial role in infrared image processing by suppressing noise and using edge detection to distinguish complex backgrounds. However, lightweight networks tend to overlook these features as the network depth increases. To address this issue adaptively, we utilized the CGA Fusion module to fuse low-level and high-level features. The adaptive strategy leverages the CGA module to enable attention information interaction, thereby enhancing the effectiveness of feature fusion.The decoupled detection head design carries a significant computational and parameter burden. To address this issue, our proposed model employs a shared convolution strategy. Simultaneously, we compensated the normalization strategy and convolution layers to prevent a decrease in accuracy. By employing GN and DEConv, we significantly augmented the expressive and generalization capabilities of feature extraction. This advancement effectively enhances the model’s ability to capture infrared details with greater precision and accuracy.ATFL [31] can decouple background and target features and handle hard samples, while PIoUv2 achieves efficient regression and fast convergence. By combining these two, the improved model demonstrates significant performance enhancements in infrared target detection, balancing detection accuracy, computational efficiency, and robustness.

## 2. Proposed Methodology

In this section, as illustrated in Figure 1, we present a comprehensive overview of the proposed LFIR-YOLO model. It includes the novel enhancements introduced in the backbone and neck architecture, the design of a lightweight detection head for improved efficiency, and the optimization of the loss function to enhance detection performance. Each of these components has been specifically designed to address the challenges inherent in infrared target detection, particularly in resource-constrained environments.

### 2.1. The C2f_DWR Module

The standard YOLOv8 model (Ultralytics, Frederick, MD, USA)frequently struggles with false positives and missed detections when processing low-contrast, low-resolution infrared images, as well as in multi-scale and multi-object scenarios, failing to meet specific application requirements. The bottleneck blocks [32] within the C2f module are a key contributor to this issue, which trades off some detection performance for faster training. Additionally, the use of fixed convolutional kernels restricts the network’s ability to adapt its receptive field, limiting its capacity for flexible feature extraction. As a result, critical information often remains incomplete, necessitating further enhancements.

Dilated convolutions [33] offer a larger receptive field and richer feature representation while preserving computational efficiency and maintaining a low parameter count, making them well-suited to addressing the challenges mentioned earlier. Although traditional multi-rate dilated convolutions are designed to capture multi-scale contextual information, they often fail to effectively learn weights at larger dilation rates, limiting their ability to extract multi-scale features comprehensively. We introduced the Dilation-wise Residual (DWR) module to overcome this limitation, enabling more effective and efficient feature extraction.

As illustrated in Figure 2, the DWR module introduces a novel two-step residual feature extraction method [34] that significantly enhances the network’s efficiency in capturing multi-scale information. The method comprises two critical steps: Region Residualization (RR) and Semantic Residualization (SR), each described in detail below.

Initially, a standard 3 × 3 convolutional layer is employed for preliminary feature extraction, followed by Batch Normalization (BN) [35] and Rectified Linear Unit (ReLU) [36] activation layers to activate distinct regional features. This process generates regional concise feature maps of varying sizes, which are subsequently forwarded as residual features to the next stage. In the second step, the regional feature maps are partitioned into multiple groups, and depth-wise dilated convolutions with varying dilation rates are applied to each group. Each channel feature in this stage utilizes a unique receptive field, effectively preventing overlap and enabling morphological filtering across regional features of different scales.

By employing two critical steps, the DWR module allows dilated convolutions to bypass the need for complex semantic information extraction. Instead, these convolutions focus on performing morphological filtering with a specific receptive field on concise regional feature maps, thereby enhancing the capture of multi-scale contextual information. The outputs from this process are aggregated, and Batch Normalization (BN) is reapplied to the combined feature maps. A final pointwise convolution then merges these features to produce the residual, which is added to the input feature map to yield a more comprehensive feature representation. Given that the original C2f module also utilizes a residual structure, it can be directly substituted with the DWR module. Furthermore, under the dilation rate settings [37], we integrate the C2f_DWR module into the higher layers of the backbone network. This strategy leverages the module’s advanced feature extraction capabilities, effectively addressing the challenges associated with infrared detection.

### 2.2. CGA Fusion Module

Attention mechanisms enhance the quality and effectiveness of feature fusion by weighting input features to highlight important information and suppress irrelevant or redundant information. Conventional feature attention modules consist of channel and spatial attention, implemented by calculating attention weights across channel and spatial dimensions. Channel attention [38] recalibrates features, while spatial attention [39] generates a Spatial Importance Map (SIM) to represent the significance of different regions. By treating different channels and pixels unequally, feature attention [40] modules aid in detecting small infrared targets and reduce false positives and negatives. Despite these advantages, spatial attention must address feature-level non-uniformity, and channel attention lacks the capability to analyze contextual information. Furthermore, these two attention mechanisms operate independently, computing weights and enhancing features without interactivity between them.

The CGA [41] mechanism addresses the issue of inter-feature information association by generating specific SIMs for each channel, as shown in Figure 3. Let X∈ℝC×H×W denote the input features. The primary objective of CGA is to generate the SIMs (i.e., W∈ℝC×H×W) corresponding to each channel. This process can be mathematically expressed using the following equations Wc and Ws:(1)Wc=c1×1(max(0,c1×1(XGAPc)))
(2)Ws=c7×7([XGAPs,XGMPs])

In the above equations, $c_{1\times 1}$ represents a convolution kernel size of 1, applied along the channel dimension. To reduce the number of parameters and limit the model complexity, the 1 × 1 convolution is first used to shrink the channels, followed by a second convolution to expand them back to the original number of channels. $c_{7\times 7}$ represents a convolution kernel size of 7. The 7 × 7 convolution can process a larger spatial area at once, thereby more effectively capturing long-range dependencies in the image features.

Subsequently, a simple addition operation is used to fuse *W_c_* and *W_s_*, resulting in the coarse SIMs: $W_{coa}=W_{c}+W_{s}$. To generate the refined SIMs W, we need to generate them, guided by the input features’ content. The process involves shuffling and interleaving each channel of $W_{coa}$ and X to reorganize them, as shown in the following equation:(3)$$W=\sigma (gc_{7\times 7}(CS([X,W_{coa}])))$$
where σ denotes the sigmoid activation function, CS(·) denotes channel shuffling, and $gc_{7\times 7}$ represents a group convolution kernel of size 7. As a result, each channel is assigned a distinct SIM, which enhances the ability to highlight relevant information within the features and focus on the characteristics of infrared targets.

The baseline model utilizes a feature fusion method involving upsampling and concatenation. However, this element-wise addition approach lacks flexibility and may not align receptive fields effectively [42]. In infrared images, low-level features typically capture edge and texture information, which is crucial for distinguishing low-contrast targets from the background, identifying objects with similar thermal characteristics, and filtering out noise. In contrast, high-level features capture semantic information and recognize complex patterns. Due to the significant differences in the encoded information and receptive fields [43] between low-level and high-level features, simple concatenation can lead to ineffective integration, resulting in potential information loss or distortion and adversely affecting detection performance. To address these challenges, we introduce a fusion scheme based on an attention mechanism—CGA Fusion.

The implemented approach involves re-fusing the low-level features from the backbone network with the features concatenated by the original model, followed by their connection to the detection head. As illustrated in Figure 4, both low-level and high-level features are first processed through the CGA module to compute spatial weights for each feature position. These weights are then used in a weighted summation method to integrate the low-level and high-level features. To enhance feature representation and address the gradient vanishing problem, skip connections are introduced during the fusion process. These connections directly incorporate the original input features into the fusion, preserving the integrity of the input information and facilitating a more direct gradient flow during training, thus simplifying the learning process. Finally, the fused features are projected through a 1×1 convolutional layer, which further integrates the feature information and reduces the channel dimension, yielding the final fused feature set $F_{fuse}$.
(4)$$F_{fuse}=c_{1\times 1}(F_{low}\cdot W+F_{high}\cdot (1-W)+F_{low}+F_{high})$$

### 2.3. Lightweight Shared Detail-Enhanced Convolution Detection Head

The baseline model incorporates three distinct detection modules, each designed to identify small, medium, and large targets at the P3, P4, and P5 scales, respectively. Each module employs a decoupled head approach, separating classification and regression tasks into two distinct computational branches. This design results in the detection head consuming approximately one-fifth of the network’s total computational resources, leading to a substantial number of parameters that complicate the deployment of the infrared detection algorithm on edge devices. To address this challenge, we propose utilizing shared convolutions, as illustrated in Figure 5.

By employing this approach, we successfully merged the three detection modules into a single detection head, significantly reducing the number of parameters. However, this merge inevitably results in decreased accuracy, necessitating compensatory measures. According to [44], Group Normalization (GN) demonstrates better performance for localization and classification tasks within the detection head. Unlike BN, which relies heavily on batch size and can suffer performance degradation when batch sizes are small [45], GN normalizes by grouping all channels and computing the mean and variance within each group, independent of batch size. Consequently, we have replaced the normalization strategy in the convolutions with GN to enhance performance and robustness.

Additionally, to enhance the capture of infrared details and further improve the accuracy of the lightweight detection head, we employed DEConv as the shared 3×3 convolution. As illustrated in Figure 6, DEConv consists of five convolutional layers: one vanilla convolution layer and four difference convolution layers arranged in parallel for feature extraction. The vanilla convolution layer captures intensity-level information, while the difference convolution layers focus on gradient-level information enhancement. Difference convolutions compute pixel differences and use kernel weights to generate feature maps, thereby improving the expressive and generalization capabilities of the convolutions. Typical variants of difference convolutions include Central Difference Convolution (CDC) and Angular Difference Convolution (ADC), which utilize rearranged kernel weights to reduce computational costs. Vertical Difference Convolution (VDC) and Horizontal Difference Convolution (HDC) incorporate traditional edge operators into the convolution layer, encoding gradient prior knowledge to learn valuable gradient information and thereby enhancing the performance and efficiency of the convolutions.

DEConv exhibits distinctive properties in convolution operations. When multiple kernels of identical size operate with the same stride and padding on the same input, summing their outputs yields the same result as summing the kernels at corresponding positions to produce the final output. This property ensures no increase in parameter count or inference time, making DEConv suitable for lightweight applications. Given the input features $F_{in}$, DEConv can produce the output $F_{out}$ with the same computational cost as a vanilla convolutional layer, achieved using the re-parameterization technique:(5)$$F_{out}=DEConv(F_{in})=\sum_{i=1}^{5}F_{in}*K_{i}=F_{in}*K_{cvt}$$
where $K_{i}(i=1:5)$ denotes the five convolution kernels, ∗ denotes the convolution operation, and $K_{cvt}$ denotes the converted kernel resulting from combining the parallel convolutions. DEConv can efficiently extract infrared detail features such as texture, shape, and temperature gradients, aligning well with the accuracy requirements of detection tasks. Finally, due to the application of shared convolution kernels, the regression task may encounter inconsistencies in target scales for each detection head. Therefore, a scale layer is added after the corresponding convolution layers to adjust the features, enhancing the stability of the model.

### 2.4. Loss Function Optimization

YOLOv8 improves upon YOLOv5 by eliminating the object loss and adopting Binary Cross Entropy (BCE) for classification loss, along with CIoU + DFL for regression loss. Additionally, YOLOv8 introduces an anchor-free, center-point-based strategy, wherein the detection model’s task is to predict the distances from the target center to the four boundaries of the bounding box.

#### 2.4.1. Classification Loss Optimization

In infrared images, background features often dominate, leading to an imbalance where the central regions of the image disproportionately influence the gradient update direction. During training, certain target features can be categorized as complex samples. The ATFL function effectively addresses this challenge by decoupling target features from the background, thereby allowing the model to focus more intensively on the target features. To leverage this capability, we have optimized BCE using the ATFL function, enhancing the model’s ability to prioritize relevant features and improve detection performance.

We can enhance the accuracy of our detection system by setting thresholds that distinguish easily recognizable backgrounds from difficult-to-recognize targets. By increasing the losses associated with targets and reducing those linked to the background, the model is able to focus more on the target features. To further improve real-time performance, an adaptive mechanism is employed. The Focal Loss (FL) function effectively addresses sample imbalance through a modulation factor ${\left(1-p_{t}\right)}^{\gamma }$. The function is defined as follows:(6)$$p_{t}=\left\{\begin{array}{c} p,\;if\;y=1\\ 1-p,\;others \end{array}\right.$$
(7)$$FL(p_{t})={\left(1-p_{t}\right)}^{\gamma }L_{BCE}$$
where p denotes the predicted probability, $L_{BCE}$ denotes the BCE function, and y denotes the truth label. The focusing parameter γ is adjusted to reduce the contribution of easily classified samples. However, the FL function can encounter difficulties when learning from complex samples. To mitigate this issue, we introduce a threshold mechanism that classifies samples with predicted probabilities above 0.5 as easy and those below this threshold as complex. Additionally, we have made adaptive improvements to the hyperparameters to optimize time and cost efficiency. The final loss function is defined as follows:(8)$$ATFL=\left\{\begin{array}{c} -{\left(\lambda -p_{t}\right)}^{-\ln(p_{t})}\log(p_{t})\;p_{t}\leq 0.5\\ -{\left(1-p_{t}\right)}^{-\ln({\hat{p}}_{c})}\log(p_{t})\;p_{t}>0.5 \end{array}\right.$$
where ${\hat{p}}_{t}$ represents the predicted value for the next epoch, $p_{t}$ represents the current average predicted probability value, λ is a fixed hyperparameter, and $-\ln(p_{t})$ and $-\ln({\hat{p}}_{t})$ are adaptively improved hyperparameters.

#### 2.4.2. Regression Loss Optimization

The following formula can summarize the existing IoU-based loss:(9)$$Loss=L_{IoU}+R(a(B,B_{gt}),b(B,B_{gt}),c(B,B_{gt}),\ldots )$$

$L_{IoU}$ is the essential IoU loss function, which treats the four boundaries of the anchor box as a whole for the regression task. B and $B_{gt}$ represent the anchor box and the ground truth boxes, respectively. a, b, c are examples of penalty factors. R is the penalty term, a function of the penalty factor. These penalty factors are typically geometric measures used to quantify the degree of matching between the anchor box and the ground truth box.

One component of the penalty term in CIoU measures the distance between the center points of the anchor box and the target box, while another component addresses differences in aspect ratios between the anchor and target boxes. However, CIoU’s penalty term primarily emphasizes maximizing the overlap area between the boxes. As a result, during the regression process, the anchor box tends to expand its size to increase this overlap, which can lead to suboptimal performance, particularly for small targets or targets with significant shape differences. Furthermore, the penalty term requires revision to better account for shape differences, as it does not adequately consider the size of the target object [46]. This limitation adversely affects the regression performance in such scenarios.

In response to these limitations, we adopted the PIoUv2 loss function, which introduces key innovations in both the penalty factor design and the application of a focus mechanism. PIoUv2 integrates a penalty factor with a gradient adjustment function that is based on the quality of the anchor box. This function directly minimizes the distances between the four edges of the anchor box and the corresponding edges of the target box, guiding the anchor box to align more precisely with the target box along a nearly straight path. This approach effectively prevents unnecessary enlargement of the anchor box, thereby enhancing the efficiency and accuracy of the regression. The size-adaptive penalty factor in PIoUv2 is defined as follows:(10)$$P=(\frac{dw_{1}}{w_{gt}}+\frac{dw_{2}}{w_{gt}}+\frac{dh_{1}}{h_{gt}}+\frac{dh_{2}}{h_{gt}})/4$$
where $dw_{1}$, $dw_{2}$, $dh_{1}$, and $dh_{2}$ represent the absolute distances between the corresponding edges of the predicted box and the target box, while $w_{gt}$ and $h_{gt}$ denote the width and height of the target box. Using P as the penalty factor in the loss function prevents the enlargement of the anchor box because its denominator includes only the parameters of the target box. Furthermore, P exhibits adaptability to the size of the target. Afterwards, the loss function achieves the focus mechanism using a non-monotonic attention function. By incorporating an attention layer, the loss function significantly enhances its ability to focus on medium- and high-quality anchor boxes. The specific implementation is as follows:(11)$$L_{PIoU}=L_{IoU}+1-e^{-P^{2}},0\leq L_{PIoU}\leq 2$$
(12)$$q=e^{-P},q\in (0,1]$$
(13)$$u(x)=3x\cdot e^{-x^{2}}$$
(14)$$L_{PIoU\_v2}=u(\lambda q)\cdot L_{PIoU}=3\cdot (\lambda q)\cdot e^{-{\left(\lambda q\right)}^{2}}\cdot L_{PIoU}$$

In the above equation, u(λq) is the attention function, where q replaces the traditional penalty factor P. The range of q is (0,1], used to quantify the quality of the anchor box. When q reaches its maximum value of 1, it signifies perfect alignment between the anchor box and the target box; at this point, the corresponding penalty factor P is 0. As P increases (indicating a lower degree of alignment between the anchor box and the target box), the value of q gradually decreases, reflecting a decline in anchor box quality.

ATFL enhances the recognition of target features, while PIoUv2 improves the accuracy of anchor box regression. Together, these functions comprehensively augment the effectiveness of infrared target detection. Additionally, PIoUv2’s efficient regression mechanism and fast convergence significantly enhance both the training and inference efficiency of the model. ATFL, through its adaptive mechanism, optimizes loss calculation and reduces computational burden. This combination not only increases the model’s overall computational efficiency but also makes it better suited for deployment in resource-constrained environments.

## 3. Experimental Analysis

Table 1 presents the experimental environment and the settings of some hyperparameters for LFIR-YOLO:

### 3.1. Datasets

To better evaluate the model’s generalization ability, we used two datasets to test the performance of the improved model—the publicly available FLIR infrared dataset released by Teledyne and the multispectral dataset from the University of Tokyo. The FLIR dataset we used was the latest v2 version. The total number of annotated frames in this version of the dataset has been expanded from 14,452 to 26,442, covering both infrared and visible light components. However, most of the images lack registration between the two modalities. The data were collected using a pair of thermal imaging and visible light cameras mounted on a vehicle. The thermal imaging camera used was the Teledyne FLIR Tau 2, with a resolution of 640 × 512 and a 13 mm focal length lens. It offers a 45-degree horizontal field of view (HFOV) and a 37-degree vertical field of view (VFOV), allowing the capture of high-resolution infrared images within a relatively wide field of view. Each frame in the dataset was sampled from entirely independent video sequences, encompassing various locations and weather conditions. We ultimately selected five target categories—person, bike, car, motor, and bus. A total of 11,329 images were selected from the original training and validation sets and were re-divided into a training set and validation set in an 8:2 ratio, resulting in 9063 images in the training set and 2266 images in the validation set. The original test set remained unchanged.

The Multispectral Dataset comprises far-infrared, mid-infrared, near-infrared, and RGB images. The dataset was collected in the University of Tokyo campus environment [47]. The far-infrared images used for training were captured by a thermal imaging camera, Nippon Avionics InfReC R500, at a rate of 1 frame per second during both day and night. The thermal imaging camera has a resolution of 640 × 480 and is capable of capturing an HFOV of 32 degrees and a VFOV of 24 degrees. To simulate real-world driving conditions, the camera was mounted on a small vehicle, with the lighting intensity set to approximate that of standard vehicle headlights. We selected 7512 far-infrared images and randomly divided them into training, validation, and test sets in an 8:1:1 ratio. This dataset includes five categories—person, car, bike, barricade, and car_stop.

For the analysis of the datasets, the FLIR dataset was specifically designed for traffic monitoring, making it well-suited for vehicle detection and object recognition in dynamic scenes. In contrast, the multispectral Ddataset primarily captures relatively static, close-range scenes, such as sidewalks and streets, with a focus on pedestrian detection and recognition.

### 3.2. Evaluation Metrics

The objective detection task evaluation metrics primarily include the model’s complexity, the number of parameters, and the computational load. Additionally, accuracy was evaluated using metrics such as Precision (P), Recall (R), and Mean Average Precision (mAP). Specifically, mAP@0.5 represents the Mean Average Precision with an Intersection over Union (IoU) threshold of 0.5. The formulas for calculating these accuracy metrics are as follows:(15)$$P=\frac{TP}{TP+FP}$$
(16)$$R=\frac{TP}{TP+FN}$$
(17)$$mAP=\frac{\sum_{i=1}^{i=M}{AP}_{i}}{M}$$

In the formulas above, TP refers to the number of correctly predicted targets in the dataset. Conversely, FP denotes the incorrectly predicted targets, where the model identifies the location of a target but assigns the wrong category. FN represents the number of missed detections where the model fails to identify the correct targets. $AP_{i}$ is the area under each category’s Precision–Recall (PR) curve; the higher the precision, the larger the area. M represents the number of categories in the dataset. The mAP is obtained by averaging the AP values of all categories.

### 3.3. Ablation Experiment

We conducted ablation experiments on the FLIR and multispectral datasets to validate the impact of each modification module on the actual detection results in the proposed LFIR-YOLO model. The specific training process is as follows: after data division, a Learning Rate Decay strategy was employed during the training process to progressively optimize model parameters. The lr0 controls the update amplitude of the model parameters, while the lrf adjusts the rate at which the learning rate decreases during training. The final learning rate is determined by the product of the initial learning rate and the decay factor. Stochastic Gradient Descent (SGD) was used as the optimizer, with a momentum of 0.937 to accelerate convergence. The training process consisted of 200 iterations, during which the learning rate gradually decreased to promote stable convergence toward the optimal solution, thereby mitigating oscillations and ensuring stabilit.

For the design of the YOLOv8 baseline model, we referenced the loss function weight settings from its application on the COCO dataset. The weight parameters for the position regression loss function were set to λ = 7.5 and μ = 1.5. In the experiments with the improved LFIR-YOLO model on the FLIR and multispectral datasets, two loss weights were adjusted. One was the modulating factor in the focal loss function, which was used to address the class imbalance problem, and was ultimately set to 1.5. λ in ATFL is a fixed hyperparameter that adjusts the sensitivity of the loss function under different thresholds; it was set to 0.25 for the experiment. The regression loss, using PIoU v2, required the introduction of a single hyperparameter λ to control the dynamic behavior of the attention mechanism. Since the hyperparameter adjustment process is relatively simple, λ = 1.3 was chosen as the optimal value to ensure effective training and optimization on both datasets. The experimental results are shown in Figure 7.

As shown in Table 2, in Experiment Group 2, the C2f_DWR module resulted in improvements in R on both the D1 and D2 datasets, with increases of 1.7% and 0.7%, respectively. The C2f_DWR module enhances the extraction of multi-scale information by broadening the receptive field, which is particularly beneficial for capturing distant objects and addressing occlusions in complex scenes. However, P experienced a slight decline on both datasets, likely due to the increased complexity introduced by the DWR module, which led to a higher incidence of false positives in certain low-contrast backgrounds. This phenomenon was particularly pronounced in the D1 dataset, where dynamic factors such as vehicle lights and reflections in complex traffic scenes increased the likelihood of false detections. Furthermore, the incorporation of the C2f_DWR module and its novel two-step residual structure contributed to a reduction in both the number of parameters and the computational load of the YOLOv8n-DWR model.

In Experiment Group 3, the CGA Fusion module demonstrated a significant enhancement in detection performance on the D1 dataset, with the Mean Average Precision (mAP@0.5) increasing to 76.0%. This improvement can be attributed to the adaptive feature fusion mechanism of the CGA module, which effectively balances low-level edge information with high-level semantic information in complex backgrounds, particularly within the densely packed and intricate FLIR dataset. The CGA module also aids in suppressing noise interference in challenging infrared backgrounds, leading to improvements in both Precision and Recall. The notable increase in mAP@0.5 on the D1 dataset indicates that the CGA Fusion module excels in multi-object detection within dynamic scenes. However, the performance on the D2 dataset exhibited a different trend. Despite the enhancements achieved through feature fusion, the mAP@0.5 experienced a decrease of 0.2%, primarily attributed to a 0.5% reduction in recall, which consequently led to an overall decline in detection accuracy. The D2 dataset consists of simpler, static scenes where the thermal contrast between objects and backgrounds is lower, with pedestrians being the main target. In such low-resolution, low-contrast scenarios, the CGA module’s feature fusion mechanism may not exhibit the same advantages it has in complex scenes. In fact, due to weaker thermal contrast in these simpler scenes, the model might struggle to accurately identify objects, resulting in a drop in recall. Furthermore, while the CGA module performs well in complex backgrounds, in simpler settings, the feature fusion may introduce unnecessary details, leading to errors in the model’s predictions.

Following the introduction of the LSDECD module, an improvement in Precision (P) and Mean Average Precision at 0.5 (mAP@0.5) is observed on the D1 dataset; however, R decreased by 0.6%. This outcome indicates that the LSDECD module effectively enhances Precision, likely by more accurately identifying key object features through its shared convolution design, thereby reducing false positive rates. Nevertheless, the decline in Recall suggests that the model may have struggled to detect all potential objects, missing some targets. This limitation may stem from the LSDECD module’s challenges in managing complex backgrounds; while it improves Precision for easily detectable objects, it may be less effective at identifying distant or occluded objects, resulting in missed detections. This phenomenon was particularly evident in the D1 dataset, which features more dynamic, distant, or complex background targets. This scenario illustrates that while the LSDECD module optimizes computational efficiency, it may inadvertently oversimplify certain complex features, particularly those that necessitate detailed processing of distant targets, ultimately leading to insufficient feature extraction and missed detections. In contrast, the LSDECD module demonstrated improvements across all metrics on the D2 dataset. This can be attributed to the generally simpler scenes in D2, where there is clearer differentiation between objects and backgrounds. The lightweight design of the LSDECD module is thus more suitable for handling such relatively straightforward scenes.

After improving the loss function, P decreased on both datasets. This phenomenon can be attributed to the fact that the improved loss function increased the model’s sensitivity when handling complex targets, leading the model to expand the detection range when encountering difficult-to-distinguish targets. Although this increased sensitivity helps detect more targets, it also raises the likelihood of false positives, particularly in complex backgrounds or low-contrast scenarios, leading to a decrease in Precision. R increased significantly on both datasets, indicating that the model became more comprehensive in detecting all potential targets. The PIoU v2 module reduced target localization errors through more precise bounding box regression, ensuring greater robustness in the model’s boundary box handling. ATFL focused on hard-to-detect targets, reducing the contribution of easily detectable objects to the loss, helping the model focus on complex or blurry targets, thus reducing missed detections. Overall, the introduction of PIoU v2 + ATFL significantly improved Recall and mAP, demonstrating stronger object detection capabilities in both complex and simple scenes. Although Precision slightly decreased, this trade-off was made to enhance recall and overall detection performance, making the model more suitable for environments where target recognition is particularly challenging.

When comparing Group 8 as a baseline with Group 6, the removal of the LSDECD module resulted in a relatively consistent R, although other performance metrics exhibited a decline. While the lightweight convolution design initially enhanced detection efficiency, its absence led to decreased computational and feature extraction efficiency. This reduction resulted in an increased number of false positives in complex backgrounds, adversely impacting P and mAP. Furthermore, the model likely lost the advantages in small target optimization provided by detailed convolution. Despite R remaining unchanged, the increase in false positives, particularly in complex scenes, suggests that the detection of small or distant targets became less accurate, thereby lowering Precision and mAP.

In comparison with Group 7, the removal of the CGA Fusion module rendered the model more lenient in detecting potential targets. Without CGA Fusion’s feature selection and enhancement capabilities, the model exhibited a tendency to be aggressive in identifying objects, including non-targets, which led to an increase in R. Moreover, since the CGA Fusion module plays an important role in minimizing false positives and optimizing feature fusion, its removal significantly reduced the model’s ability to differentiate between noise and background. This resulted in a substantial increase in false positives, thereby significantly lowering Precision. More category information can be found in Table 3.

### 3.4. Discussion on the Performance of the Model in Infrared Scenarios

To analyze the practical detection capabilities of the LFIR-YOLO model in different scenarios, several representative infrared detection cases were selected from the two datasets for further analysis and validation of the experimental results. These cases include typical multi-object dynamic scenes on urban roads and static scenes, as well as special cases such as low-contrast scenes, complex background scenes, long-distance target scenes, and edge-occluded target scenes.

In Figure 8a, vehicles are distributed at varying distances, representing a typical multi-object detection scene. Additionally, the vehicle in the middle is a distant target. The model successfully detected multiple vehicles and provided high confidence scores, demonstrating good performance in recognizing vehicle targets. The fact that the model can accurately detect both near and far objects indicates that its ability to handle multi-scale targets has been enhanced. Similarly, in Figure 8b, the vehicle in the foreground appears to be dynamically blurred, suggesting that the vehicle is in motion. The model was able to detect the target with high confidence, indicating that the improved loss function endows the model with robustness when dealing with dynamic targets. The enhanced feature extraction module may have contributed to the model’s ability to detect dynamically blurred objects. Moreover, Figure 8c represents a low-contrast outdoor urban scene where the detection system must capture and recognize distant pedestrian targets despite the small temperature differences. However, the model showed lower confidence in detection, suggesting that while the improved model can detect targets, it still has limitations in distinguishing distant targets with weak thermal radiation in low-contrast environments. In the future, data augmentation strategies could be employed to increase training data for low-contrast targets, or thermal source-based enhancement strategies could be developed, which would assign greater weight to thermal intensity and help the model better handle specific challenges in low-contrast scenarios.

Figure 9a depicts a static low-contrast open scene with an occluded vehicle on the right. However, the improved model failed to detect this target, indicating that the training dataset may lack sufficient samples of edge targets, resulting in weaker detection performance for such cases. Additionally, the feature extraction for edge targets may not be sufficiently robust, suggesting an area for future improvement. Figure 9b presents a complex urban traffic scene characterized by numerous static elements—such as buildings—in the background, alongside multiple vehicles and pedestrians. In this context, the detection system must accurately distinguish between various objects within the complex background. The CGA Fusion module, through its adaptive feature fusion strategy, integrates low-level edge details with high-level semantic information. This fusion mechanism enhances the model’s ability to capture low-level edge details while simultaneously leveraging high-level semantics to differentiate between targets and background, effectively suppressing background noise. The results demonstrate that all visible vehicles and pedestrians were successfully detected, indicating the model’s capacity to maintain balanced and efficient performance in complex environments.

### 3.5. Performance Comparison

To comprehensively analyze the object detection capabilities of the LFIR-YOLO model, we performed training and testing using the publicly available FLIR and multispectral datasets [47]. Faster R-CNN, as a classical two-stage detection algorithm, has achieved good performance in visible image detection. However, due to the low contrast and unique characteristics of infrared images, these models often perform poorly in infrared image detection. We selected Faster R-CNN as a benchmark to demonstrate the advantages of single-stage models in terms of computational complexity and accuracy. The selection of SSD as a comparative model aims to demonstrate the performance of early single-stage detection architectures in managing multi-scale infrared image targets while simultaneously emphasizing the performance gap between these models and the YOLO series. YOLOv3-tiny, YOLOv5n, and YOLOv6n represent mid-stage, lightweight, single-stage detection models that are extensively utilized in edge computing and real-time applications due to their efficiency and adaptability. However, in infrared image detection, these models are often constrained by computational resources or insufficient optimization for infrared features. YOLOv9-tiny and YOLOv10-n are the latest lightweight detection models, and although they have been optimized in terms of parameter count, their performance in infrared images remains limited due to their inadequate adaptation to infrared features. We selected these models for comparison to demonstrate the further optimization of LFIR-YOLO in infrared image detection. YOLO-IR and YOLO-FIRI, specifically designed for infrared imagery, serve as benchmarks for evaluating LFIR-YOLO’s improvements. This comparison emphasizes the model’s superior performance in terms of both lightweight architecture and detection accuracy, showcasing its enhanced computational efficiency and target detection capabilities.

In the experiment, the Faster R-CNN model used ResNet-50 as the backbone, as it provides a good balance between depth, feature extraction capability, and computational complexity. The learning rate, optimizer, and batch size are consistent, as shown in Table 1, with the RPN (Region Proposal Network) positive-to-negative sample ratio set to 1:3. The anchor scales were set to [128, 256, 512] and the anchor ratios were set to [0.5, 1.0, 2.0], with a total of 100 training epochs. For the SSD model, the input image size was set to 512 × 512, with an initial learning rate of 0.01, and a learning rate decay strategy was applied. The momentum was set to 0.9 to accelerate training and avoid oscillation. Six anchor boxes were used for target detection at different scales and aspect ratios, with a batch size of 16, and training was conducted for 200 epochs. The hyperparameters for the YOLO series were consistent, as shown in Table 1, to minimize the impact of hyperparameter settings on model performance. The comparative analysis of the experimental results is detailed in Table 4.

In the context of two-stage detection algorithms, Faster R-CNN exhibits poor performance in mAP for infrared image object detection. Likewise, the SSD series of single-stage detection algorithms have larger parameter counts and computational loads, but their detection performance on both datasets is not significantly different from that of the two-stage algorithms. However, the subsequent YOLO series of single-stage detection algorithms have significantly surpassed the aforementioned early detection algorithms in terms of performance. Earlier YOLO models, including YOLOv3-tiny and YOLOv6-n, not only have high parameter counts and computational demands but also perform poorly in terms of mAP. While YOLOv5-n demonstrates reasonable accuracy, it is still hindered by a significant computational load. The latest lightweight models, YOLOv9-tiny and YOLOv10-n, similarly suffer from high computational demands and share identical drawbacks to YOLOv5-n. These models show varying degrees of incompatibility with infrared image object detection. Although YOLO-IR and YOLO-FIRI are YOLO variants specifically designed for infrared images, experimental results indicate that they remain less effective than our model in terms of lightweight performance and mAP.

### 3.6. Model Lightweighting Analysis

To comprehensively highlight the lightweight characteristics of the proposed model, this paper provides a detailed analysis of the computational complexity associated with each component of the network, as illustrated in Figure 10. This quantification not only highlights the overall efficiency improvements achieved but also offers a detailed comparison against the baseline model, clearly demonstrating the advantages of the proposed modifications.

A key innovation lies in the C2f_DWR module, integrated within the backbone network, which significantly contributes to reducing the computational burden. Specifically, this module achieves a reduction of 0.07G and 0.03G in computational cost, showcasing its effectiveness in optimizing the processing pipeline without sacrificing the model’s capacity to capture and process critical features. This efficiency gain is particularly noteworthy as it highlights the module’s ability to enhance the model’s performance while maintaining a lightweight structure.

Furthermore, the introduction of the CGA Fusion module for advanced feature fusion exemplifies a balanced approach to network enhancement. Although this module comprises three additional layers, it results in only a minimal increase of 0.29G in computational complexity. This increment is strategically contained within acceptable limits, ensuring that the model remains lightweight while benefiting from improved feature integration. The careful design of the CGA Fusion module ensures that it contributes positively to the model’s overall performance without imposing excessive computational demands.

Additionally, the novel detection head, carefully designed to optimize performance, plays a crucial role in further reducing the model’s computational complexity. This reduction is not merely incremental but rather a significant contribution that enhances the model’s overall efficiency. The streamlined detection head design enables the model to perform more effectively in real-world scenarios, where computational resources may be limited and faster inference times are paramount.

Together, these innovations result in a model that excels in balancing high performance with low computational cost, making it particularly well-suited for applications where both efficiency and accuracy are critical. The enhancements introduced, including the C2f_DWR module, CGA Fusion module, and the innovative detection head, collectively reinforce the model’s lightweight nature, positioning it as a state-of-the-art solution in the domain of efficient neural network architectures.

### 3.7. Real-Time Performance Evaluation

To verify the real-time performance of the LFIR-YOLO model, we conducted a comprehensive evaluation of its detection efficiency on various hardware platforms. The results are presented in Table 5. The real-time performance metrics were measured using average inference time (latency) and frames per second (FPS). The specific testing procedure is as follows: first, we selected the target hardware device (either GPU or CPU) and deployed the optimized model onto it, generating a set of randomly sampled input data. To ensure optimal performance during inference, we performed 200 warmup iterations prior to the formal testing phase. These warmup iterations allowed the model to be fully loaded and optimized. Subsequently, we conducted the formal inference tests, running 1000 inference iterations in a loop and recording the time for each iteration. After the tests, the average inference time was calculated based on the recorded time array, and FPS was determined as the inverse of the average inference time.

In autonomous driving scenarios, the requirements for latency are particularly stringent, with target latencies generally needing to be less than 50 milliseconds to ensure sufficient reaction time. As shown in Table 5, all GPU devices met this requirement, with excellent FPS performance. For CPU devices, however, only those with a performance exceeding that of the AMD EPYC 7453 series exhibited satisfactory real-time performance. For instance, the Gold 6130 CPU demonstrated lower performance, with an FPS of only 12.2, which may be sufficient for low- or medium-speed video streams in tasks such as surveillance or traffic monitoring. However, for high-speed scenarios, higher-performing hardware is clearly necessary to further improve inference speed.

Overall, the LFIR-YOLO model performs exceptionally well on GPU devices, achieving very high FPS, making it well-suited for applications with strict latency requirements, such as autonomous driving. While performance on CPU devices has improved, it still presents certain limitations for scenarios requiring high real-time capabilities, reinforcing the model’s strengths in GPU-based environments.

### 3.8. Detection Results Visualization Comparison

Figure 11 and Figure 12 provide comparative visualizations of representative images from the FLIR and multispectral datasets. Specifically, the images on the left are processed using the YOLOv8n model, while those on the right are processed using the enhanced LFIR-YOLO model. In Figure 11a, which depicts a typical multi-scale infrared target scene, the baseline YOLOv8n model demonstrates three missed detections in the person category and one in the car category. Conversely, Figure 11b illustrates an occlusion scene where the YOLOv8n model fails to detect two instances of the person category. Figure 12a shows an infrared low-contrast scene where our proposed model achieves improved detection accuracy compared to the baseline model. In contrast, Figure 12b highlights a case of false detection by the baseline model, which inaccurately identifies an additional car at the edge of the image.

### 3.9. Analysis of Object Detection Results in Dynamic Scenarios

In our experiment, we conducted a detailed analysis of object movement across different scenarios to evaluate the model’s performance in complex dynamic environments. By conducting experiments on three representative cases, we demonstrated the model’s robustness and detection accuracy in tracking moving objects. The scenarios included a regular traffic road environment (Figure 13a), a pedestrian walkway environment under very low light at night (Figure 13b), and a high-speed road environment with strong light conditions (Figure 13c). The scenario images are derived from the FLIR dataset, and the data for each scenario consist of a series of continuous frames captured at 30 FPS, which fully demonstrate the test intensity and real-time performance of dynamic object detection. We selected 30 image frames captured within one second as visualization results to illustrate the model’s detection performance with moving objects, and the results are presented in Figure 14, Figure 15 and Figure 16. The test results show that, in all three complex scenarios, the detection confidence of the model reaches approximately 90% for moving objects, indicating stable recognition capability in various real-world applications.

Additionally, the detection performance could be affected by several factors when the object is in rapid motion, mainly including relative motion blur and system latency. However, since our model has a high processing speed and the image capture time is within milliseconds, the movement of objects at normal traffic speeds does not significantly impact detection accuracy. Moreover, our system is capable of effectively handling dynamic changes through optimized detector latency. Nevertheless, in the case of extremely high-speed movement, the detection accuracy may slightly decline due to the rapid relative movement of the object to the camera, but overall, the system’s detection capability remains practical.

### 3.10. Generalization Experiment

To thoroughly assess the generalization capability of the LFIR-YOLO model, we employed the RGB images from the multispectral dataset as the test dataset for our generalization experiments. Specifically, the RGB images were captured using a Logicool HD Pro Webcam c920r, and the dataset consists of a total of 7512 images, with 3740 taken during the day and 3772 taken at night, with a resolution of 640 × 480. To ensure the reliability and comparability of the experimental results, we adhered strictly to the experimental setup described in Section 3.5. This included maintaining the same hyperparameter settings, training batches, number of epochs, and dataset-splitting strategy. As shown in Table 6, the improved model continues to demonstrate a significant advantage in detection accuracy, indicating that our proposed LFIR-YOLO model exhibits excellent generalization performance across different datasets, effectively maintaining high detection accuracy and stability when handling diverse types of images.

## 4. Conclusions

Existing deep learning-based models for infrared object detection often face challenges in balancing model size and detection accuracy, resulting in poor performance across diverse scenarios. To address these limitations, we proposed a novel lightweight model called LFIR-YOLO. Our approach integrates several innovations to enhance detection capabilities while maintaining computational efficiency. We incorporated the DWR module into the high-level C2f component of the backbone network, expanding the receptive field without augmenting the parameter count. This enhancement facilitates the network’s ability to harness contextual information for the accurate detection of low-contrast targets and improves its capacity to interpret multi-scale infrared features. Furthermore, we introduced a content-guided attention mechanism within the neck network, which adaptively modulates the feature fusion ratio. This mechanism ensures comprehensive interaction among feature representations and effectively utilizes low-level features to attenuate image noise, thereby optimizing object detection performance.

To further enhance the model’s balance between efficiency and accuracy, we reconfigured the detection head to use a shared convolution approach and refined the loss function by replacing CIoU Loss and BCE Loss with PIoUv2 Loss and ATFL. These modifications not only accelerate convergence but also significantly improve target detection and localization accuracy. Empirical results demonstrate that our proposed LFIR-YOLO model outperforms the baseline YOLOv8n model, achieving average accuracies of 78.9% and 87.2% on two infrared image datasets. Additionally, the model exhibits reductions in computational load and parameter count by 4.2G and 0.49 × 10^6^, respectively. These advancements represent a substantial improvement in balancing accuracy and model reduction. In future work, we will focus on the deployment of LFIR-YOLO on edge devices to further extend its applicability and real-world effectiveness.

## Figures and Tables

**Figure 1 sensors-24-06609-f001:**
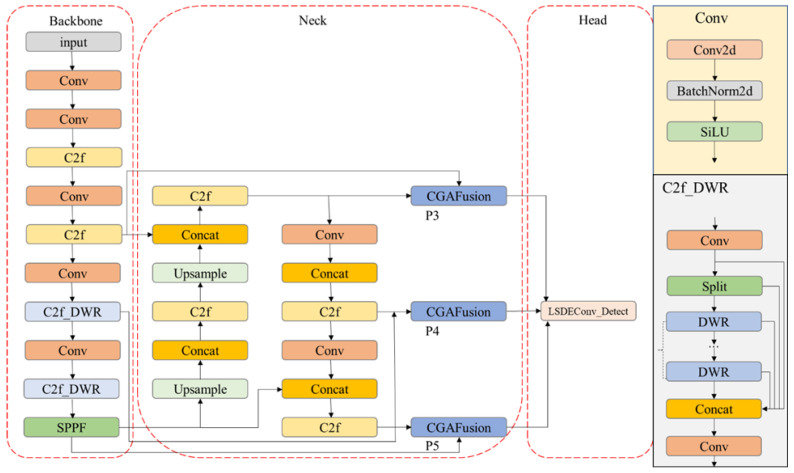
LFIR-YOLO model structure diagram.

**Figure 2 sensors-24-06609-f002:**
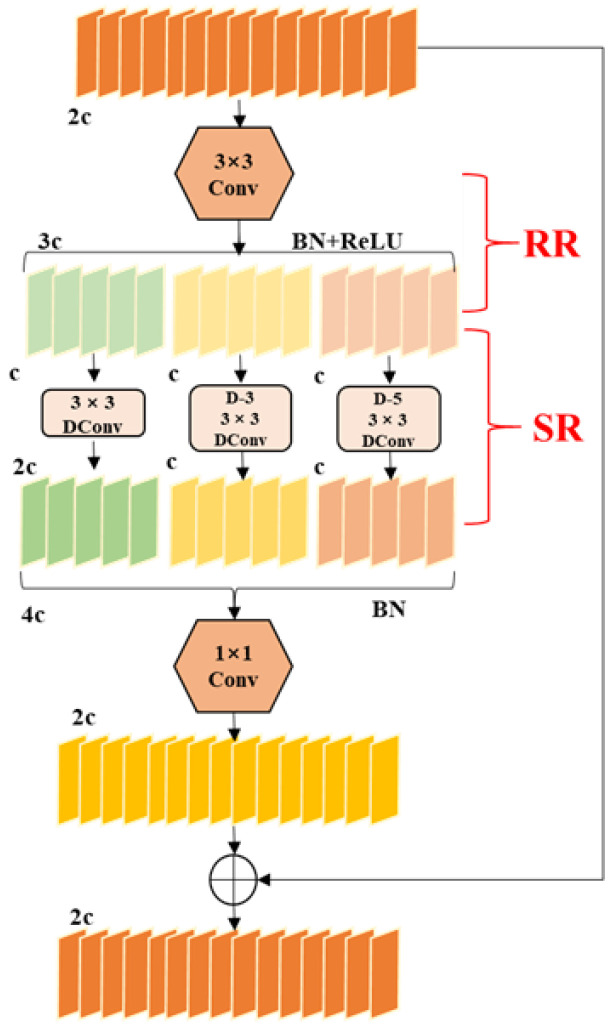
Dilation-wise Residual module.

**Figure 3 sensors-24-06609-f003:**
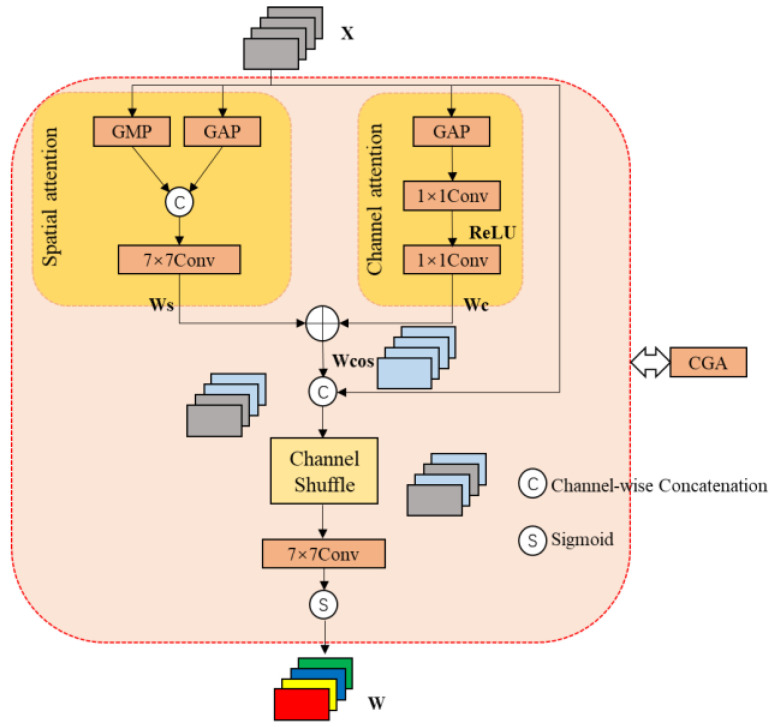
Content−guided Attention module.

**Figure 4 sensors-24-06609-f004:**
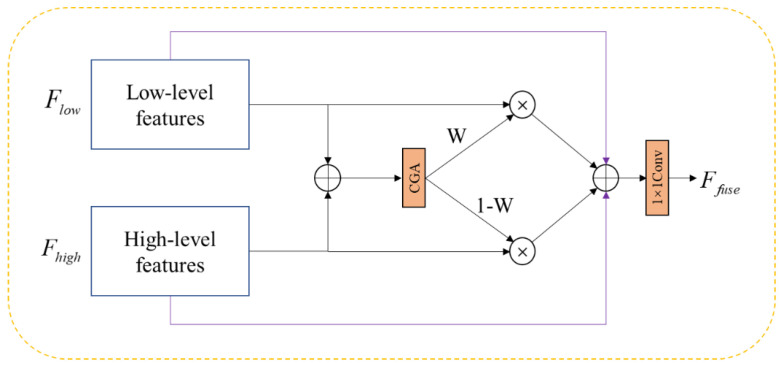
Content−guided Attention Fusion module.

**Figure 5 sensors-24-06609-f005:**
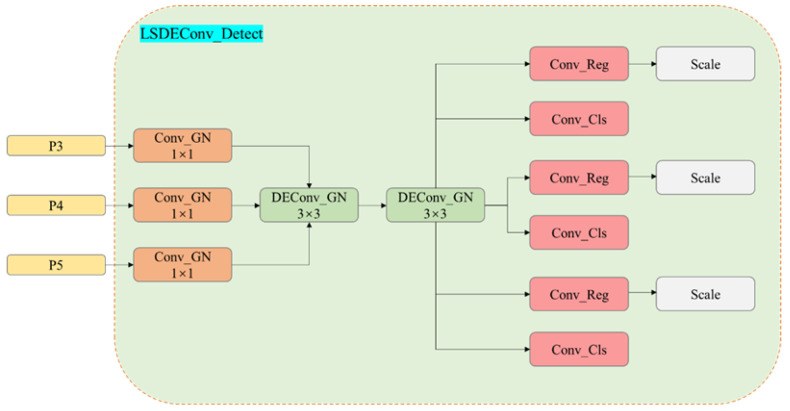
Lightweight Shared Detail-enhanced Convolution Detection Head structure diagram.

**Figure 6 sensors-24-06609-f006:**
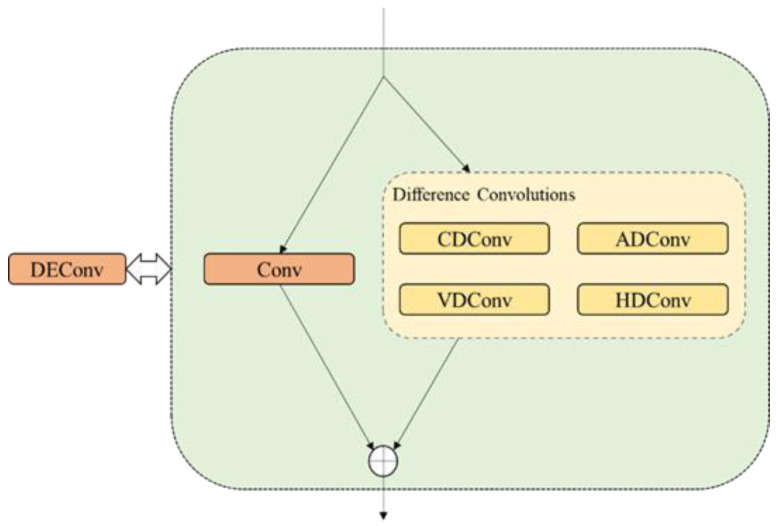
Details of DEConv.

**Figure 7 sensors-24-06609-f007:**
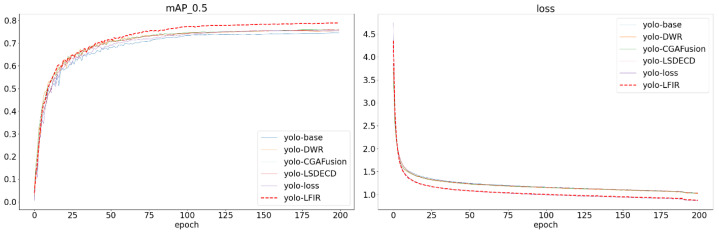
Ablation experiment comparison chart for mAP@0.5 and box_loss.

**Figure 8 sensors-24-06609-f008:**
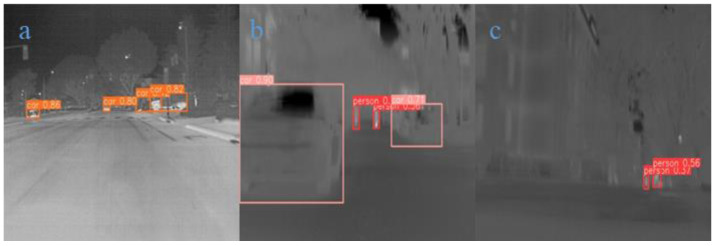
Random infrared example Group 1. (**a**): A multi-object detection scene with vehicles at varying distances. (**b**): A dynamic blur detection scene where the vehicle in the foreground is in motion. (**c**): A low-contrast outdoor urban scene focused on detecting distant pedestrians.

**Figure 9 sensors-24-06609-f009:**
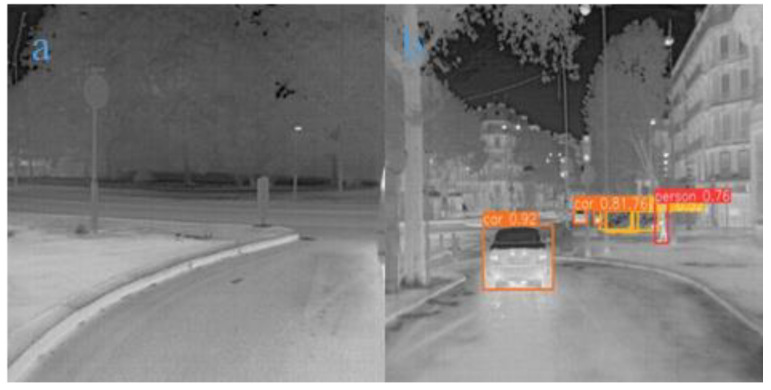
Random infrared example Group 2.

**Figure 10 sensors-24-06609-f010:**
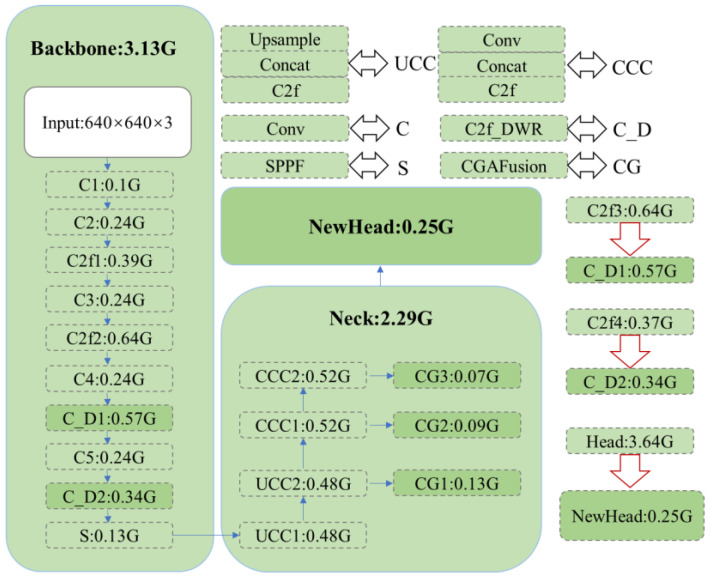
Computational complexity of the model.

**Figure 11 sensors-24-06609-f011:**
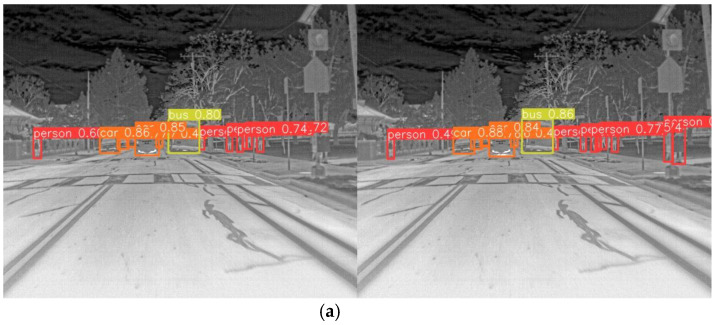
(**a**) FLIR image detection results for multi-scale target scene. (**b**) FLIR image detection results for occlusion scene.

**Figure 12 sensors-24-06609-f012:**
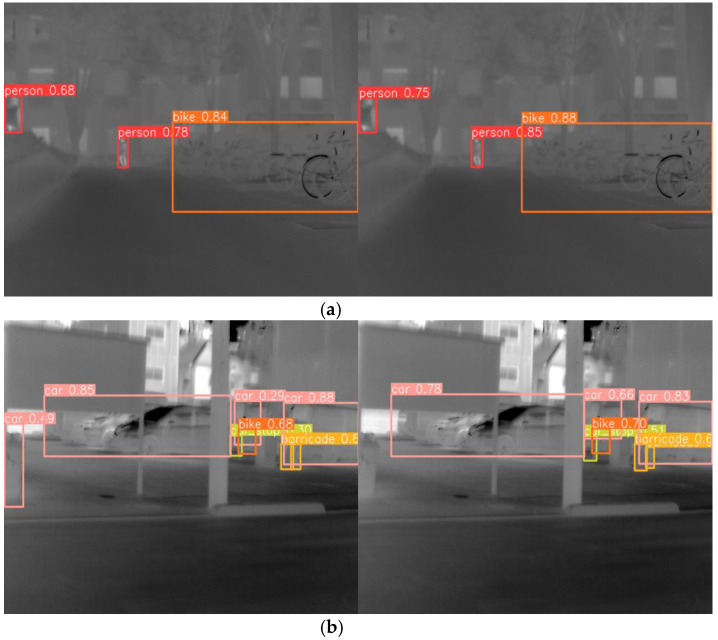
(**a**) Multispectral image detection results for infrared low-contrast scene. (**b**) Multispectral image detection results for false detection case.

**Figure 13 sensors-24-06609-f013:**
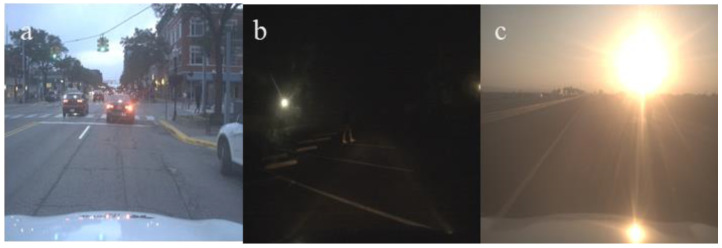
Representative scenarios for dynamic object detection. The scenarios include a regular traffic road environment (**a**), a pedestrian walkway environment under very low light at night (**b**), and a high-speed road environment with strong light conditions (**c**).

**Figure 14 sensors-24-06609-f014:**
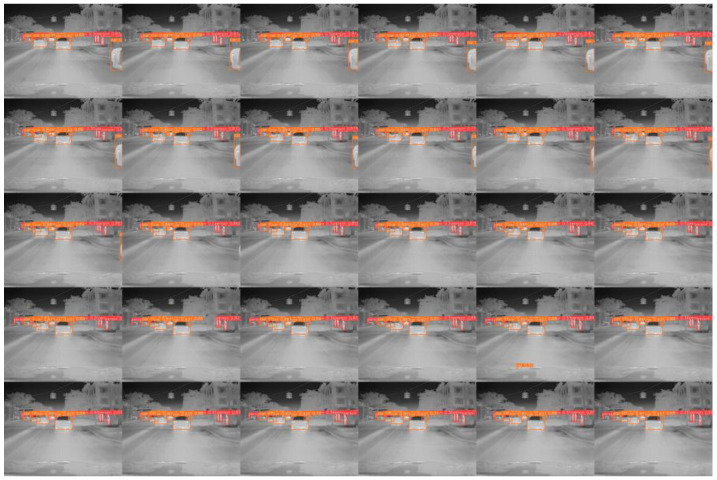
Regular traffic road environment.

**Figure 15 sensors-24-06609-f015:**
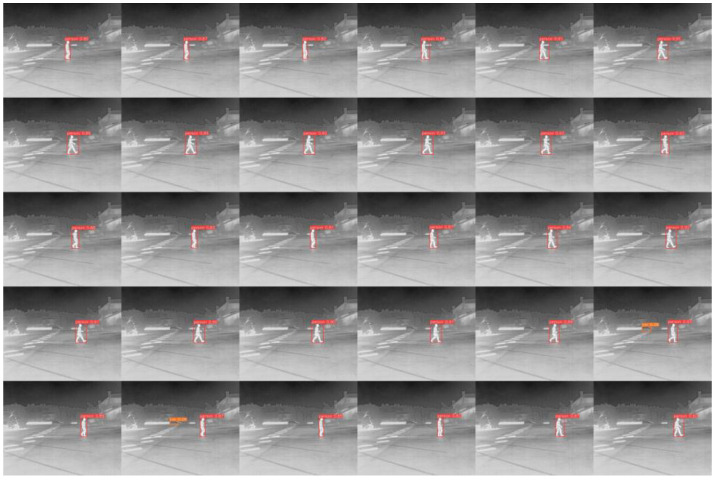
Pedestrian walkway environment under very low light at night.

**Figure 16 sensors-24-06609-f016:**
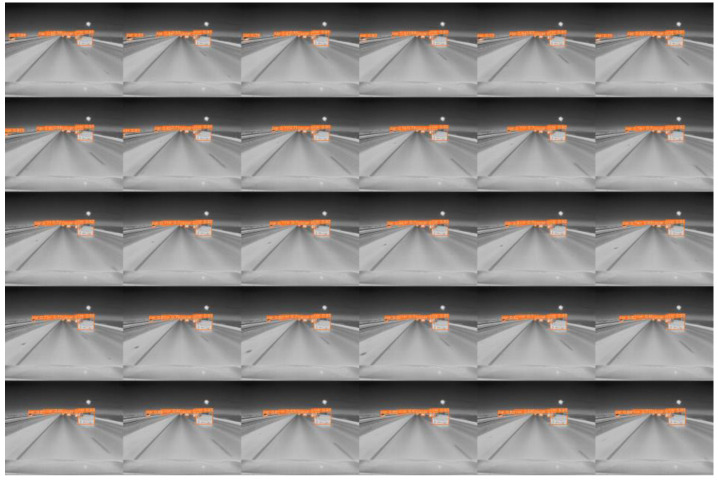
High-speed road environment with strong lighting.

**Table 1 sensors-24-06609-t001:** Experimental environment configuration.

Project	Environment	Hyperparameter	Setting
OS	Ubuntu 22.04	Input Resolution	640 × 640
CPU	Gold 5418Y	Initial Learning Rate 0	0.01
GPU	RTX 4090	Learning Rate Float	0.01
CUDA	12.1	Batch Size	16
Pytorch	2.3.1	Workers	8
Python	3.11.4	Epochs	200

**Table 2 sensors-24-06609-t002:** Ablation experiment results.

G	D	C	L	P	P/%		R/%		mAP@0.5/%		mAP@0.5:0.95/%		Params/10^6^	GFLOPs/G
					D1	D2	D1	D2	D1	D2	D1	D2		
①	―	―	―	―	81.5	85.1	65.3	78.5	74.6	84.6	45.5	51.8	3.15	8.9
②	√	―	―	―	81.0	84.7	67.0	79.2	75.5	84.8	47.2	52.4	3.09	8.8
③	―	√	―	―	82.5	86.0	66.1	78.0	76.0	84.4	47.7	52.1	3.31	9.2
④	―	―	√	―	80.9	85.5	67.0	79.6	76.0	84.8	47.4	52.3	2.56	5.5
⑤	―	―	―	√	80.9	84.5	66.6	79.4	75.6	85.3	46.2	52.4	3.15	8.9
⑥	√	√	―	√	82.2	86.4	69.0	80.5	77.8	86.4	48.3	53.1	3.24	9.1
⑦	√	―	√	√	82.4	85.2	**69.3**	**81.2**	78.3	86.6	48.3	53.0	**2.50**	**5.4**
⑧	√	√	√	√	**83.7**	**87.7**	68.8	80.4	**78.9**	**87.2**	**49.4**	**53.9**	2.66	5.7

G: Groups, D: DWR, C: CGAFusion, L: LSDECD, P: PIoUv2+ATFL, D1: FLIR, D2: Multispectral.

**Table 3 sensors-24-06609-t003:** Detection accuracy for each category.

G	D	C	L	P	AP (%)									
					D1					D2				
					Person	Bike	Car	Motor	Bus	Person	Car	Bike	Barricade	Car_Stop
①	―	―	―	―	78.8	67.0	81.7	66.3	79.1	89.0	90.3	81.2	81.8	80.6
②	√	―	―	―	80.3	68.5	82.3	67.0	79.4	90.3	89.3	81.8	81.1	81.4
③	―	√	―	―	80.5	67.8	82.4	68.4	80.7	89.7	88.9	81.4	80.7	80.3
④	―	―	√	―	80.8	69.3	82.7	67.9	79.1	89.3	89.9	83.6	80.3	81.7
⑤	―	―	―	√	79.4	67.9	82.6	68.3	79.6	90.3	90.3	83.4	80.6	82.6
⑥	√	√	―	√	81.4	66.8	83.1	76.2	81.8	90.9	90.2	85.7	82.8	82.0
⑦	√	―	√	√	80.9	67.8	**83.2**	76.1	**83.3**	91.3	89.9	**86.3**	82.8	83.1
⑧	√	√	√	√	**81.8**	**70.9**	**83.2**	**76.9**	81.8	**91.6**	**93.1**	85.2	**82.9**	**83.2**

**Table 4 sensors-24-06609-t004:** Comparison experiment results.

Base Model	Params/10^6^	GFLOPs/G	mAP@0.5/%		mAP@0.5:0.95/%	
			D1	D2	D1	D2
Faster-RCNN	15.8	28.3	54.0	61.6	32.3	36.4
SSD	18.9	35.3	55.4	60.7	33.5	36.5
YOLOv3-tiny	12.17	19.1	65.6	79.0	38.1	47.2
YOLOv5-n	**2.65**	7.8	75.2	84.4	45.9	51.2
YOLOv6-n	4.50	13.1	71.6	81.8	43.3	49.5
YOLOv9-tiny	2.66	11.0	72.1	81.2	43.5	49.0
YOLOv10-n	2.70	8.4	73.0	83.6	43.8	50.3
YOLO-FIRI [25]	7.2	20.4	76.0	83.5	47.5	50.0
YOLO-IR [26]	7.3	9.1	76.7	85.0	48.6	51.9
Ours	2.66	**5.7**	**78.9**	**87.2**	**49.4**	**53.9**

**Table 5 sensors-24-06609-t005:** Inference time comparison across different devices.

GPU	Latency/ms	FPS	CPU	Latency/ms	FPS
RTX 4060 8 GB	12.36	80.9	Gold 6130	82.22	12.2
RTX 3060 12 GB	12.29	81.4	E5-2680 v4	71.71	13.9
RTX 2080ti	10.17	98.3	Gold 6330	51.45	19.4
V100	9.96	100.4	Processor(Skylake)	50.45	19.8
RTX 3080	9.67	103.5	AMD EPYC 7453	46.70	21.4
A100	9.55	104.7	Platinum 8352V	45.60	21.9
vGPU-32 GB	8.34	119.9	Platinum 8255C	43.79	22.8
V100-SXM2	7.84	127.5	AMD EPYC 9K84	41.77	23.9
A40	7.60	131.6	Gold 6348	34.05	29.4
RTX 3090	7.48	133.7	Platinum 8457C	**27.45**	**36.4**
A800	6.17	162.2			
L20	5.16	193.8			
RTX 4090	5.14	194.7			
L40	**4.93**	**203.0**			

**Table 6 sensors-24-06609-t006:** Results of the generalization experiment.

Base Model	Params/10^6^	GFLOPs/G	mAP@0.5	mAP@0.5:0.95
Faster-RCNN	15.8	28.3	54.7	32.9
SSD	18.9	35.3	57.5	35.0
YOLOv3-tiny	12.17	19.1	71.4	41.6
YOLOv5-n	**2.65**	7.8	79.2	45.9
YOLOv6-n	4.50	13.1	76.4	44.4
YOLOv8-n	3.15	8.9	80.0	46.8
YOLOv9-tiny	2.66	11.0	78.4	45.2
YOLOv10-n	2.70	8.4	79.7	45.2
Ours	2.66	**5.7**	**81.0**	**47.2**

## Data Availability

Access to the FLIR dataset: https://www.flir.com/oem/adas/adas-dataset-form/#anchor29 (accessed on 11 October 2024). Access to the Multispectral dataset: https://drive.google.com/drive/folders/1YtEMiUC8sC0iL9rONNv96n5jWuIsWrVY (accessed on 11 October 2024). Access to the code: https://github.com/Brise02/LFIR-Yolo.git (accessed on 11 October 2024).

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
