# Peer review of "LFIR-YOLO: Lightweight Model for Infrared Vehicle and Pedestrian Detection"

_sensors, 2024, doi:10.3390/s24206609_

Round 1
Reviewer 1 Report
Comments and Suggestions for Authors
To address the problems of insufficient color information, texture details and low spatial resolution in infrared images, this paper, a lightweight model called LFIR-YOLO is proposed. The model is based on the YOLOv8 architecture, aiming to improve the accuracy of infrared vehicle and pedestrian detection in night traffic scenarios to meet the requirements of practical deployment through the enhanced feature extraction and fusion capabilities, and the optimized loss function.
Lines 45-51, in the introduction section, although relevant studies of YOLO series models and infrared imaging are reviewed, the practical application scenarios and challenges of infrared imaging detection are relatively brief. It is suggested to supplement some cases or data of infrared imaging in practice to enhance the actual background correlation of the problem.
Line 407-409, the experiment setting is not clear, although listed the experimental environment and some super parameter setting, but did not detail the specific steps of the experiment, such as data pretreatment, model training specific strategy (such as learning rate adjustment strategy), model training weight and whether the cross validation, suggest the corresponding part.
In lines 412-432, the experimental section mentions multiple indicators (mAP, P, R), but in-depth discussion of these results is lacking. For example, although the results of accuracy and recall rate are given, it is not analyzed in depth why some improvement modules improve more significantly in some cases and less in some cases. More interpretation of the experimental results and analysis of possible reasons or limitations are recommended.
In lines 435-445, although the article illustrates the experimental results, an independent discussion section is missing. A part of the discussion can be added to focus on the performance of the model in different scenarios, as well as possible limitations, such as in some infrared scenes, the model still has false or underreporting reasons. A more in-depth analysis of experimental results including performing the model in specific situations and possible direction of improvement is suggested.
Rines 450-455, inadequate model comparisons, and in comparison experiments, although compared with some existing models, do not provide sufficient information to explain why these models were chosen as the benchmark for comparison, and the specific configurations and parameter settings of these models.
Lines 497-502, although the lightweight model improvement is mentioned in the paper, the actual performance of LFIR-YOLO on different hardware platforms (such as embedded devices or GPU servers), especially the balance between detection speed and accuracy. This helps the reader to understand the actual performance of the model in different application environments.
Code and data are not available, and supplementary material is mentioned in the paper, but no links to the code and datasets are provided, which limits the transparency and verifiability of the study. In the current research setting, open codes and data sets are important to facilitate scientific progress and community collaboration. It is recommended to provide access links to the code and datasets so that other investigators can reproduce and validate the experimental results.
Reviewer 2 Report
Comments and Suggestions for Authors
Adding the following may improve your work.
- Page 5: “In these equations, k x k denotes the size of the convolution kernel, max (0, x) denotes the ReLU activation function, and [ .] denotes channel-wise concatenation. “There is no k here in the equations. What is the need for generalizing if constants are used in the equations?
- Page 11 “. Since many categories in the dataset have too few labeled samples, we removed categories with fewer than 500 samples and ultimately selected five target categories: person, bike, car, motor, and bus.” This is not the right strategy to train and test the robustness of the model. It would help if you came up with a solution, such as using class imbalance loss or another strategy rather than removing the class categories.
- It is better to provide details about the two datasets, such as the weather conditions and characteristics of infrared sensors. While these datasets cover typical scenarios, they may not represent all possible conditions for infrared imaging, such as varying weather conditions or different infrared sensor characteristics.
- It is better to include mAP in addition to mAP@0.5.
- How scalable is this model for other datasets besides infrared? It should not be limited to infrared datasets.
Reviewer 3 Report
Comments and Suggestions for Authors
Done

Round 2
Reviewer 2 Report
Comments and Suggestions for Authors
Authors addressed my concerns.
Author Response
Thank you for your positive feedback. We are glad that the revisions addressed your concerns. We appreciate your valuable comments, which helped us improve the quality of the manuscript.